# Diagnostic and Prognostic Characteristics of Circulating Free DNA Methylation Detected by the Electrochemical Method in Malignant Tumors

**DOI:** 10.3390/cancers13040664

**Published:** 2021-02-07

**Authors:** Li-Yue Sun, Zi-Ming Du, Yu-Ying Liu, Yan-Hong Li, Xiao-Min Liu, Ting Wang, Jian-Yong Shao

**Affiliations:** 1State Key Laboratory of Oncology in South China, Guangzhou 510060, China; sunly@sysucc.org.cn (L.-Y.S.); duzm1@sysucc.org.cn (Z.-M.D.); liuyy@sysucc.org.cn (Y.-Y.L.); liyanh@sysucc.org.cn (Y.-H.L.); liuxm@sysucc.org.cn (X.-M.L.); tt091251@163.com (T.W.); 2Collaborative Innovation Centre for Cancer Medicine, Guangzhou 510060, China; 3Department of Molecular Diagnostics, Sun Yat-Sen University Cancer Centre, Guangzhou 510060, China; 4Department of Cancer Prevention, Sun Yat-Sen University Cancer Centre, Guangzhou 510060, China; 5Department of Clinical Laboratory, Sun Yat-Sen University Cancer Centre, Guangzhou 510060, China

**Keywords:** electrochemical, circulating free DNA methylation, malignant tumor, diagnostic, prognostic

## Abstract

**Simple Summary:**

Previous studies have established an electrochemical detection method for the rapid detection of cfDNA (circulating free DNA) methylation and found that this technology might be a potential method for cancer diagnosis. However, the underlying mechanism and its role in the diagnosis and prognosis of malignant tumors are not well-characterized. In present study, we utilized the electrochemical detection method to detect the DNA methylation status by using electron microscopies and infrared spectroscopy and found that DNA with different methylated levels adsorbed to the gold surface differently, which was likely mediated by hydrophobic bonds. In addition, after detection of the cfDNA methylation status from 505 normal individuals, 725 cancer patients before treatment, and 549 patients after treatment, we found that the cfDNA adsorption rate could be used as an indicator for the diagnosis and prognosis prediction of pan-cancer. Our research provides a novel method for the liquid biopsy of cancer for diagnosis and prognosis predictions.

**Abstract:**

Prior research has established an electrochemical method based on the differential adsorption capacity of gold surfaces with different methylated DNA degrees and found that this method might be valuable for cancer diagnosis by detecting circulating free DNA methylation. However, further investigation on the underlying mechanism and validation of its diagnostic and prognostic values in a large cohort of malignant tumors was limited. We found that DNA with different methylation levels formed particles of diverse sizes on the gold surface. Hydrophobic bonds played a significant role in the binding process of methylated DNA to the gold surface. The detection condition of an adsorption time of 10 min and temperature of 20 °C was optimal. In a large cohort of plasma samples from the patients with different malignant tumors, as well as normal individuals, we found that the electrochemical detection method based on the differential adsorption capacity of methylated DNA degree on a gold surface could be used as a noninvasive tool for malignant tumor diagnosis and prognostic evaluation. The diagnostic efficiency of this method in malignant tumors was even slightly better than that of the current tumor biomarkers widely used in routine clinical practice (circulating free DNA (cfDNA) vs. carcinoembryonic antigen (CEA), 0.8131 vs. 0.7191 and cfDNA vs. CA19-9, 0.7687 vs. 0.6693).

## 1. Introduction

Malignant tumors were estimated to cause 2814.2-thousand deaths in China in 2015 [1]. Previous studies have demonstrated that an early diagnosis and screening can significantly reduce the mortality of malignant tumors [2,3,4,5]. At present, a histological examination of tissue is still the gold standard for the diagnosis of malignant tumors. Compared with a tissue examination, a blood test has many advantages, such as noninvasiveness [6], real-time detection [7], long-term monitoring [8] and fast detection [9]. However, the commonly used tumor biomarkers such as serum carcinoembryonic antigen (CEA) [10], alpha fetoprotein (AFP) [11] and carbohydrate antigens [12] have disadvantages such as poor sensitivity and specificity. In addition, imaging examinations are mainly applied to assess the efficacy of anticancer therapy. However, such examinations are expensive, but not fast, and need the judgment of radiologists, and moreover, some local hospitals lack the relevant equipment [13,14]. Some traditional tumor biomarkers are used in clinical postoperative recurrence monitoring and the efficacy evaluation of malignant tumor patients, but their sensitivity and specificity are insufficient [15,16]. Therefore, it is urgent to find novel and accurate tumor biomarkers for the diagnosis, curative effect verification, recurrence monitoring and prognosis evaluation of malignant tumors.

Liquid biopsy markers, including circulating free DNA (cfDNA)/circulating tumor DNA (ctDNA), play an important role in the early diagnosis, efficacy monitoring and prognostic evaluation of malignant tumors [17]. Abnormal hypermethylation of the gene promoter region is a unique epigenetic characteristic of malignant tumors, and it is also an important clinical biomarker [18,19]. The main clinical detection methods for cfDNA/ctDNA are PCR-based detection methods, including methylation-specific PCR [20], methylation quantitative PCR [21], digital droplet PCR [22] and high-throughput sequencing technology [23]. Compared with those PCR-based detection methods, the electrochemical methods have the advantages of being fast, sensitive, economical and simple and, also, have the potential to be further transformed into point-of-care testing [24]. Among the numerous electrochemical detection methods for cfDNA methylation, the method of directly adsorbing DNA on a gold surface does not require modification of the recognition layer, which can reduce the detection cost and improve the detection efficiency [24]. Therefore, the electrochemical detection method that directly uses a gold surface to adsorb DNA is a potential detection method for cfDNA methylation detection. 

Recent studies have developed a set of electrochemical detection systems called eMethylsorb based on the different adsorptions between different DNA bases and gold surfaces. Nevertheless, this method still requires sulfite conversion for DNA methylation detection [25]. Later, they discovered that different methylation levels of DNA had differential adsorption capacities to gold surfaces and further developed an electrochemical detection method that does not require sulfite conversion [26]. In a small cohort of 100 patients with stage IV colorectal cancer (CRC), stage IV breast cancer (BC) patients and 45 normal people, it was found that this method had good diagnostic efficiency for CRC and BC. However, this study had the following shortcomings. First, the mechanism of the differential adsorption capacity of DNA with different methylation levels on the gold surface was unclear. Second, there was a lack of pan-cancer types and different tumor-node-metastasis (TNM) stages of malignant tumor patients to verify the technical method. Third, there was a lack of research on the efficacy for malignant tumor diagnosis, monitoring and prognostic evaluation. Therefore, the clinical application of the technology still needs further investigation. 

The present study intends to establish an electrochemical detection technology based on the differential adsorption capacities of different methylation levels DNA on a gold surface. The underlying mechanism, as well as the values of the diagnosis and prognosis predictions of the technology, are also investigated. 

## 2. Results

### 2.1. Differential Adsorption Capacity of DNA with Different Methylation Levels on Gold Surface

First of all, we investigated the physical morphology and possible mechanism of DNA methylation at different methylation levels, cancer DNA and normal tissue DNA adsorbed to a gold surface. In order to verify the results of a previous study [26], we investigated the physical morphology of methylated DNA and non-methylated DNA in liquid through transmission electron microscope (TEM). Methylated DNA showed a highly condensed distribution, while non-methylated DNA was fragmented. Abnormal methylation is considered to be a unique epigenetic feature of malignant tumors. Therefore, we also compared the difference in the condensed morphology between the tumor DNA and normal DNA. Compared with normal tissue DNA, tumor tissue DNA had a denser distribution and fewer gaps (Figure 1A). As shown in previous studies [26], DNA with different methylation levels show different polycondensation capabilities, and cancer DNA and normal DNA also have differences. Methylated DNA has a significantly larger coverage area (20.32% ± 5.78% vs. 5.69% ± 3.37%, *p* = 0.019) and higher DNA aggregates (13.78 ± 2.29 nm vs. 5.67 ± 3.29 nm, *p* < 0.001) than non-methylated DNA adsorption on a gold surface. Tumor tissue DNA also formed larger and higher clusters on the gold surface compared with normal tissue DNA (relative coverage area,13.51% ± 1.02% vs. 9.80% ± 1.09%, *p* = 0.013 and height, 18.07 ± 4.87 nm vs. 13.55 ± 3.92 nm, *p* = 0.029) (Figure 1B,C). The above results showed that methylated DNA and tumor DNA can form larger polycondensed particles on a gold surface, so the electrochemical system that relies on gold as a working electrode can detect the difference caused by DNA methylation. It was found that non-methylated DNA and methylated DNA showed absorption peaks at 3346 cm^−1^ and 1637 cm^−1^, respectively, through Fourier-transform infrared spectroscopy (FTIR). A group analysis illustrated that the absorption peaks were H_2_O (Figure 1D). Taken together, there are differences in the adsorption of DNA with different methylation levels on a gold surface, and there are also differences in the adsorption of tumor and normal DNA with gold. This difference can be used to establish a detection method for malignant tumors.

### 2.2. Establishment of Electrochemical DNA Methylation Detection System Based on Differential Pulse Voltammetry (DPV)

We established an electrochemical differential pulse voltammetry (DPV) mode DNA methylation detection system based on a three-electrode system and determined its optimal detection condition. The cyclic voltammetry (CV) curve showed that the system had good repeatability after running 100 times (Figure 2A). The peak current of the CV curve decreased after the adsorption of DNA compared to the baseline (non-DNA) (Figure 2B). This suggests that the CV current will change after the gold electrode adsorbs the DNA. However, this current signal change is too small, and further improvement is still needed in accurate detection, such as cfDNA detection. In order to establish a more sensitive detection method, the potential range of the DPV curve was determined by the CV curve (Figure 2C). The peak current of the DPV curve became significantly lower after the adsorption of DNA (Figure 2D). Since the interference of the background current signal is eliminated, the DPV curve changes more obviously after the gold electrode adsorbed the DNA compared with the CV curve. According to previous studies [26], this change in the current peak value can reflect the amount of DNA adsorbed on the electrode. It was found that the adsorption rate of DNA was highest at the DNA concentration of 10 ng/μL (Figure 2E and Appendix A), the incubation temperature of 20 °C (Figure 2F and Appendix A) and the incubation time of 10 minutes (Figure 2G and Appendix A) compared with different conditions. In addition, the current adsorption rate of methylated DNA was significantly higher than the non-methylated DNA (methylated DNA vs. non-methylated DNA, 26.00% ± 3.38% vs. 10.10% ± 3.22%, *p* < 0.05; Figure 2H). As the proportion of methylated DNA increased, its current adsorption rate also increased (Figure 2H). After shearing the DNA off tumor and normal tissues by sonication, it was found that the sheared DNA from the tumor tissue had a higher absorption rate than the original DNA samples (sheared DNA vs. non-sheared DNA, 30.37% ± 2.17% vs. 25.44% ± 3.06%, *p* = 0.019). However, there was no significant difference in the normal tissue between sheared DNA and non-sheared DNA (sheared DNA vs. non-sheared DNA, 19.53% ± 2.64% vs. 17.36% ± 2.36%, *p* = 0.207) (Appendix A).

### 2.3. Detection of cfDNA Methylation Status in Malignant Tumors Patients Using DPV-Based Electrochemical DNA Methylation Detection System

We used the DPV-based electrochemical DNA methylation detection system mentioned above to detect the cfDNA methylation status from 505 cases of normal people and 725 cases of initial diagnosed cancer patients, including 47 lung cancer (LC), 166 hematological tumor, 173 CRC, 42 gastric cancer (GC), 22 nasopharyngeal cancer, 82 HCC, 59 esophageal cancer (EC) and 134 other malignant tumors (please see Appendix A for detailed information). The results showed that the cfDNA adsorption rate in each cancer type was higher than that of normal people (*p* < 0.001) (Figure 3A and Appendix A). The cfDNA adsorption rates in CRC and HCC patients were significantly higher than that in normal people (CRC vs. normal, 31.85% ± 7.44% vs. 23.47% ± 6.54%, *p* < 0.001 and HCC vs. normal, 35.94% ± 12.17% vs. 23.47% ± 6.54%, *p* < 0.001). The area under the curve (AUC) of the CRC and HCC were 0.8245 and 0.8388, respectively (Figure 3B,C). To compare the consistence between the cfDNA and tissue DNA, we also detected the DNA adsorption rate from the tissue of CRC and HCC and found that the DNA adsorption rates of CRC and HCC tissues were also significantly higher than that of the corresponding normal tissues (CRC tissue vs. normal tissue, 31.18% ± 7.39% vs. 20.28% ± 4.39%, *p* < 0.001 and HCC tissue vs. normal tissue, 29.33% ± 3.84% vs. 19.86% ± 6.06%, *p* < 0.001). The AUC of the CRC tissue and HCC tissue were 0.9625 and 0.9050, respectively (Figure 3D,E and Appendix A). The results showed that there was a difference in the cfDNA adsorption rate between tumor patients and normal people, which could be used as a potential diagnostic marker for malignant tumors. We also tested the DNA adsorption rate in both cfDNA and DNA from the corresponding peripheral white blood cells (WBC) from the same cancer patient. The results showed that the cfDNA adsorption rate was significantly higher than the WBC DNA adsorption rate, which further proved that the method could be used to detect DNA methylation characteristics derived from tumors in cfDNA (Appendix A).

### 2.4. Effect of cfDNA Adsorption Rate in Malignant Tumour Diagnosis, Curative Effect Monitoring and Prognostic Evaluation

In order to verify the effects of this method in the diagnosis, recurrence monitoring and prognosis of malignant tumors besides in 725 initial diagnosed malignant tumors patients and 505 normal people, we also tested 409 cancer patients with response, and 140 cancer patients with relapse/progress, after treatment. The clinical characteristics of the patients are listed in Table 1. The results showed that the absorption rate of cancer cfDNA was significantly higher than that of normal people (32.56% ± 9.39% vs. 23.47% ± 6.54%, *p* < 0.001) (Figure 4A). The AUC of the receiver operating characteristic (ROC) was 0.8125 (Figure 4B and Appendix A). When the current adsorption rate cut-off value was 28.38%, the diagnostic sensitivity of the cfDNA adsorption rate in the malignant tumor diagnosis was 76.83%, the specificity was 80.99%, the positive predictive value was 84.81%, the negative predictive value was 68.39% and the accuracy rate was 76.83%. The subgroup analysis showed that there was no significant difference in the adsorption rate between patient genders and ages (Appendix A). The cfDNA adsorption rate of patients with relapse/progress was significantly higher than patients with response after treatment (32.65% ± 9.44% vs. 26.75% ± 9.90%, *p* < 0.001) (Figure 4A and Appendix A). The AUC of the ROC was 0.6847 (Figure 4B). When the current adsorption rate cut-off value was 28.38%, the sensitivity of the cfDNA was 67.14%, the specificity was 63.08%, the positive predictive value was 38.37%, the negative predictive value was 84.87% and the accuracy rate was 64.12%.

In addition, we also analyzed the role of the cfDNA current adsorption on the prognosis of 28 patients with stage IV LC and 65 patients with stage IV CRC, respectively. Using x-tile software, the cut-off values of the current absorption rates in LC and CRC were 27.30% and 30.40%, respectively. With the cut-off values, there were 21 patients with low adsorption rates and seven patients with high adsorption rates in LC, respectively. There were 21 patients with low adsorption rates and 44 patients with high adsorption rates in CRC, respectively. In LC and CRC patients, the median follow-up time was 6.73 (1.12–15.74) months and 8.61 (0.2–23.79) months, respectively. Among LC patients, the median progression-free survival (PFS) of patients with a low adsorption rate was 3.29 ± 0.39 months, which was significantly lower than that of patients with a high adsorption rate (*p* = 0.004). Among patients with CRC, the PFS of patients with a high adsorption rate was also significantly superior to that of patients with a low adsorption rate (high adsorption rate CRC vs. low adsorption rate CRC: 12.29 ± 2.21 vs. 7.89 ± 0.98, *p* = 0.001) (Figure 4C).

### 2.5. Comparison of cfDNA Adsorption Rate and Serum CEA, CA19-9 and AFP Levels in Malignant Tumors

To further assess the effect of the cfDNA adsorption rate in the diagnosis of cancer, we compared the cfDNA adsorption rate and serum tumor biomarkers such as CEA, CA19-9 and AFP in the diagnosis of malignant tumors. The results showed that, compared with serum CEA levels, the AUC of the cfDNA adsorption rate in CRC, LC, GC, and EC was better than CEA (CRC cfDNA vs. CEA, 0.7980 vs. 0.7936, LC cfDNA vs. CEA, 0.7773 vs. 0.6922, GC cfDNA vs. CEA, 0.6788 vs. 0.6211 and EC cfDNA vs. CEA, 0.8033 vs. 0.5918). In CRC and GC, the cfDNA adsorption rate was also superior to the serum CA19-9 level (CRC cfDNA vs. CA19-9, 0.7938 vs. 0.6889 and GC cfDNA vs. CA19-9, 0.6572 vs. 0.5821). However, in HCC, the AUC of the serum AFP level was better than the cfDNA adsorption rate (HCC cfDNA vs. AFP, 0.7681 vs. 0.8719) (Figure 5 and Appendix A).

## 3. Discussion

On the basis of previous research, we established a DNA methylation detection system based on electrochemical methods. This method can convert differences in DNA methylation levels into electrical signals, so that methylation differences in DNA can be detected quickly and sensitively. Since DNA methylation plays an important role in the occurrence and development of malignant tumors, we also found that this method can sensitively diagnosis tumor patients from normal people in tissue DNAs and plasma cfDNAs. This study also found that the cfDNA adsorption rate can be used as an indicator for pan-cancer diagnosis, curative effect monitoring and prognostic evaluation. This detection system is expected to provide a novel method for the clinical application of malignant tumor liquid biopsy.

Our research found that methylated DNA and non-methylated DNA had different condensed forms in a liquid environment. In addition, methylated DNA formed larger and higher clusters on the gold surface compared to non-methylated DNA. There are similar differences between tumor tissue DNA and normal tissue DNA. Previous studies have depicted that the sequence, structure and intermolecular forces of DNA affect the adsorption state on the gold surface, but the mechanism of the differential binding of different methylation levels of DNA is still unclear [27,28,29,30,31,32]. Basic research has shown that CpG island methylation not only affects the gene expression, but also affects the DNA structure, which depends on the DNA sequence and the location of methylation sites [33,34]. Derreumaux et al. found that CpG methylation can affect the kinetics of DNA, which manifests as a decreased flexibility of the DNA [35]. Therefore, CpG methylation can significantly change the structure and physical properties of DNA. Sina et al. found that DNA with different methylation levels were differentially adsorbed to the gold surface but did not investigate its mechanism [26]. We found that H_2_O had differences in the adsorption of methylated DNA and non-methylated DNA to the gold surface, suggested that hydrophobic bonds were involved in the differential adsorption of DNA with different methylation levels on the gold surface. Previous studies have confirmed that hydrophobic bonds affect the solvation of DNA molecules, as it is one of the main factors that affect the binding of DNA to gold surfaces, especially in a liquid environment [28]. DNA bases usually hold H_2_O molecules in specific locations, such as the base nitrogen ring, outer ring amino groups and ketone groups [36]. Hydrophobic methylated groups appear in the DNA sequence after methylation, thereby removing the water molecules contained in these parts. Previous studies have found that DNA adsorption on a gold surface can be enhanced after the dehydration of DNA bases and a phosphate backbone [28]. Therefore, water molecules and hydrophobic bonds participate in the adsorption process of methylated DNA and the gold surface in the liquid environment. However, this hypothesis cannot fully explain the mechanism by which the current adsorption rate of cancer-derived DNA is higher than methylated DNA. Beside the hydrophobic bond, whether there are other factors affecting the adsorption of methylated DNA on a gold surface remains to be further studied.

We measured the stability of the detection system by CV and DPV measurements, then found that, when the DNA concentration was 10 ng/μL and the incubation time was 10 min, the adsorption rate of DNA was the highest, which is consistent with previous studies [26]. We also found that the adsorption rate of DNA was highest when the ambient temperature was 20 °C. Prior studies have also suggested that ambient temperature is important for electrochemical detection. In addition to affecting the rate of electrode reaction, temperature also affects the transfer process of relevant chemical substances to the electrode surface [37,38]. This also provides a prerequisite for the better application of technology. After the tissue DNA was interrupted by ultrasound, we also found that the DNA adsorption rate after the interruption was slightly higher than before the interruption, suggesting that the detection method can also be used for short DNA fragments, such as cfDNA.

The results showed that the method could distinguish cancer tissue DNA from normal tissue DNA more accurately through the current adsorption rate, and it can also distinguish the cfDNA of tumor patients from normal human cfDNA. The diagnostic performance of this method on tissue DNA was similar to previous studies, but the diagnostic performance of cfDNA was worse than previous studies [26]. We considered the following reasons. First, the cfDNA detected in the study by Sina et al. was derived from patients with stage IV BC and CRC. Our study included TNM stage I-IV patients and more than 15 diverse cancer types. However, the subtype analysis showed that TNM staging was not statistically different in the analysis of the CRC and HCC cfDNA adsorption rates (Appendix A). Secondly, there were differences in the initial input of the plasma. The cfDNA adsorption rate of patients with malignant tumors studied by Sina et al. was significantly higher than our study (52.88% ± 16.95% vs. 32.56% ± 9.39%), but the cfDNA adsorption rate of normal specimens was similar (23.38% ± 16.8% vs. 23.47% ± 6.54%), suggesting that there may be differences in malignant tumor specimens. The plasma initial input in previous studies was 1 mL, but it was 200 μL in our study. Prior research has shown that, in the peripheral blood of patients with advanced malignant tumors, the content of DNA per milliliter of plasma is about 17 ng [39]. Our research on the most suitable concentration also exhibited that increasing the concentration of DNA within a certain range can increase the adsorption rate of DNA. Therefore, a higher initial plasma input can increase the concentration and total amount of cfDNA, which may increase the detection efficiency. In addition, there were differences in the DNA solutions. Used as the solution in previous studies was 5×SSC, but the DNA solvent used in our study was a DNA Elution Buffer with Tris and EDTA. However, we did not see a significant difference in the adsorption rate between the two studies in tissue DNA testing. Therefore, whether it can enhance the adsorption rate in cfDNA remains to be further studied. 

We also compared the cfDNA adsorption rate and tumor biomarkers such as CEA, AFP and CA19-9 in the diagnosis of CRC, LC, HCC and other malignant tumors. Compared with CEA and CA19-9, the cfDNA adsorption rate showed a better diagnostic performance. However, the cfDNA adsorption rate showed a poor diagnostic efficiency compared with AFP in the diagnosis of HCC. The possible reason is the advanced stage of most of the included cases. Previous studies have shown that AFP levels in HCC are positively correlated with TNM staging [40]. The proportion of locally advanced or distant metastases (stage III and IV) of HCC patients was 75.00% in this study. In addition, we found that the sensitivity of the cfDNA adsorption rate (66.67%, 14/21) was better than that of the AFP levels (38.10%, 8/21) in the stage I/II HCC patients. Therefore, the diagnostic efficiency of AFP being superior to the cfDNA adsorption rate may be caused by more patients in the advanced TNM stage in this study.

The results showed that the cfDNA adsorption rate is an indicator of possible potential malignant tumor efficacy monitoring and prognosis evaluation. However, its sensitivity and specificity in monitoring efficacy is low, and its recurrence diagnostic efficacy is similar to clinical tumor biomarkers such as CEA and AFP [41,42]. When the sensitivity increases, the specificity is significantly reduced. The possible reason is that the sizes of some early recurrence lesions are too small and below the detection limit of CT/MRI. At the same time, these early recurrences sites may still release tumor biomarkers, including cfDNA [43]. This makes the current clinically dependent imaging criteria for tumor recurrence/progression delayed. Therefore, these patients need to be followed up for a long time to observe whether there will be a tumor recurrence or disease progression in follow-up studies.

However, this study still has the following problems: (1) The extraction quality and initial volume of plasma cfDNA has a greater impact on the final results. (2) The sensitivity of the system’s detection results need improvement. (3) There is a lack of prospective multicenter clinical studies to further verify its effect in the diagnosis, efficacy monitoring and prognosis evaluation of malignant tumors. Therefore, the electrochemical method for detecting the methylation adsorption rate of cfDNA needs to be further studied and perfected, and at the same time, there needs to be an in-depth study of its role in clinical malignant tumor diagnosis, curative effect monitoring and prognostic evaluation.

## 4. Materials and Methods

### 4.1. Patients Selected

Tumor tissue specimens and normal tissue specimens were obtained from patients with CRC and hepatocellular cancer (HCC) who were diagnosed at the Sun Yat-Sen University Cancer Centre from January 2018 to January 2019. Tumor patient plasmas were obtained from the remaining plasma samples of clinical testing of the Sun Yat-Sen University Cancer Centre from November 2017 to December 2019. Normal people plasma was obtained from the remaining plasma samples of clinical testing of the Department of Cancer Prevention. All patients with malignant tumors were confirmed through pathological examination. Normal people underwent a full physical examination, including, at least, chest X-rays, abdominal ultrasound and serum tumor biomarkers, and no significant evidence of tumors were found. This study was approved by the Ethics Committee of Sun Yat-Sen University Cancer Centre (no. B2017-079-01) and obtained informed consent from each person. 

### 4.2. DNA Sample Prepared

Human methylated and non-methylated DNA sets were purchased from Zymo (Zymo Research Inc, Irvine, CA, USA). The formalin-fixed paraffin-embedded (FFPE) tissue specimens were outlined by two independent reviewing pathologists and micro-dissected to separate tumor and normal tissues. Tissue specimens were first treated with xylene to remove the paraffin for DNA extraction. After, they were washed with ethanol and digested with Proteinase K overnight. Finally, we performed DNA extraction according to the instructions of the QIAamp DNA FFPE Tissue Kit (QIAGEN, Hildren, Germany). Circulating free DNA was extracted in an automated liquid processing workstation (AMTK, Beijing, China) by a magnetic bead extraction kit (DA0620 DAAN Gene, Guangzhou, China). The initial extraction volume of each plasma sample was 200 µL. cfDNA was extracted from the plasma following the standard procedures.

### 4.3. Electrochemical Detection

All electrochemical experiments were carried out using a CH1060C potentiostat (Chinstruments, Shanghai, China) with a three-electrode system consisting of a gold working electrode (2 mm in diameter), Pt counter electrode and Ag/AgCl reference electrode (Gaoss Union, Tianjin, China). Cyclic voltammetry (CV) and differential pulse voltammetry (DPV) experiments were carried out in 10-mM phosphate buffered solution containing 2.5-mM K_3_Fe (CN)_6_ and 2.5-mM K_4_Fe (CN)_6_ (Aladdin, Shanghai, China). CV experiment parameters included Init E = 0.5 V, High E = 0.5 V, Low E = −0.5 V, Scan Rate = 0.1 V/s, Sample Interval = 0.001 V, Quiet time = 2 s and Sensitivity = 1 × 10^−4^ A/V. DPV experiment parameters included Init E = −0.1 V, Final E = 0.5 V, Incr E = 0.005 V, Amplitude = 0.05 V, Pulse Width = 0.05 s, Sampling Width = 0.05 s, Pulse Period = 0.1 s, Quiet Time = 2 s and Sensitivity = 1 × 10^−5^ A/V. For DNA detection, the gold electrodes were initially cleaned by an electrode polishing machine carrying aluminum polishing powder (Gaoss Union, Tianjin, China). After washing away the residual aluminum powder in deionized water, it was dried under the flow of nitrogen. We then measured the DPV signal of the clean electrode in the electrolyte to obtain the baseline current. Then, the purified DNA (5 µL of a 10-ng/µL concentration in Elution Buffer) was adsorbed on the gold electrode for 10 minutes to get a sample current. Adsorbed DNA reduced the amount of current passing through the naked electrode, and the decrease was related to the amount of adsorbed DNA [25,26]. The relative adsorption currents due to the adsorption of the DNA samples were then measured by using an equation [26] (1): Adsorption current(%i_r_) = [(i _baseline_ − i _sample_)/i _baseline_] × 100(1)

### 4.4. TEM Measurements

Diluted human methylated DNA, non-methylated DNA, CRC tissue DNA and normal intestinal mucosal epithelial tissue DNA by 10 ng/uL were stained on the carbon-based resin of copper meshes and then stained with a saturated uranyl acetate and lead solution, and then, we observed the DNA condensed morphology on a transmission electron microscope (TEM; Hitachi H-7650, Hitachi High-Technologies Corporation, Tokyo, Japan).

### 4.5. SEM Measurements

Diluted human methylated DNA, non-methylated DNA, CRC tissue DNA and normal intestinal mucosal epithelial tissue DNA to 10 ng/µL were adsorbed on the surface of the gold electrode (Gaoss Union, Tianjin, China). After drying, we sprayed gold on the surface and observed the morphological changes between different DNAs under a scanning electron microscope (SEM; Hitachi SU8020, Hitachi High-Technologies Corporation, Tokyo, Japan) and analyzed the relative coverage area of DNA on the surface of the gold plate electrode under 100,000× magnification using Image Pro Plus 6.0 software (Media Cybernetics, Rockville, MD, USA).

### 4.6. Atomic Force Microscope (AFM) Measurements

Diluted human methylated DNA, non-methylated DNA, CRC tissue DNA and normal intestinal mucosal epithelial tissue DNA by 10 ng/uL were adsorbed on the surface of the gold electrode (Gaoss Union, Tianjin, China). After drying, we observed the morphological changes between different DNAs under an atomic force microscope (AFM; Bruker Dimension Icon, Bruker Corporation, Billerica, MA, USA) and analyzed the relative coverage height of the DNA on the surface of the gold disk electrode at 115,000× magnification using NanoScope Analysis 1.8 software (Bruker Corporation, Billerica, MA, USA). 

### 4.7. FTIR Measurements

Human methylated DNA and non-methylated DNA were diluted to 10 ng/µL and adsorbed on the surface of the gold electrode (Gaoss Union, Tianjin, China). After drying, a covalent bond connection of methylated DNA and non-methylated DNA to the surface of the gold electrode under Fourier-transform infrared spectroscopy was observed (Thermo Scientific, Nicolet IS10, Fremont, CA, USA).

### 4.8. Follow-Up

According to the National Comprehensive Cancer Network guidelines, a follow-up of patients with malignant tumors includes imaging examinations such as CT and MRI. Judging the recurrence or progression of malignant tumors mainly relies on CT, MRI and clinical medical records. The primary study endpoint is progression-free survival (PFS).

### 4.9. Statistical

All statistical analyses were performed using SPSS 20.0 (SPSS inc., Chicago, IL, USA). Categorical data was compared using the chi-square test. Continuous data was compared with the *t*-test. One-way analysis of variance, followed by Holm-Sidak, was performed for multiple group comparisons. The receiver operating characteristic curve (ROC) was used to calculate the area under the curve (AUC), sensitivity, specificity, positive predictive values, negative predictive values and accuracy rate. Survival analysis used Kaplan-Meier to compare the prognosis of the two groups. *p* < 0.05 was statistically significant.

## 5. Conclusions

Our research provides a novel DNA methylation detection method that can sensitively detect differences in overall DNA methylation within 10 min, so as to diagnose malignant tumors, monitor the therapeutic effects and evaluate the prognosis. This study verified the role of this method in the diagnosis of malignant tumors in more stages, cancer types and specimens and, also, initially evaluated its role in efficacy monitoring and prognosis evaluations. It is expected to provide a new method for liquid biopsies of malignant tumors.

## Figures and Tables

**Figure 1 cancers-13-00664-f001:**
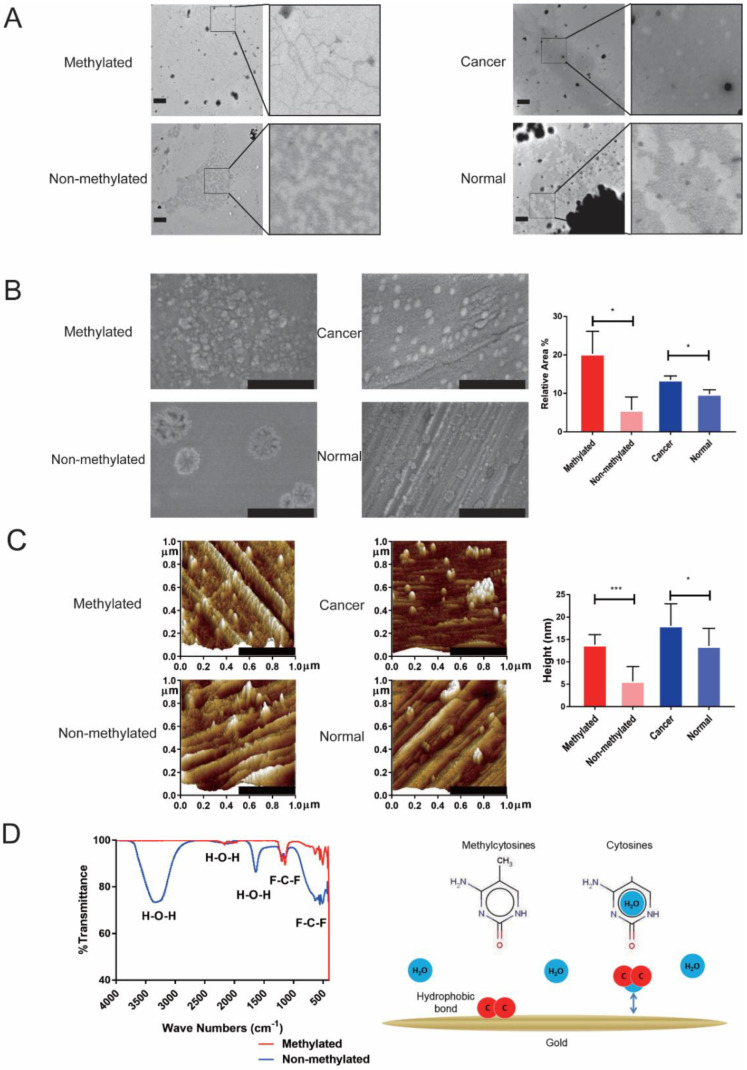
Methylation status affects the physical characteristics of DNA absorbed on a gold surface. (**A**) TEM images show methylated DNA and non-methylated DNA with different forms in a liquid environment. DNA from tumor tissue and normal tissue also show different forms (scale bar: 600 nm). (**B**) SEM images show methylated DNA has a larger adsorption area on the gold surface than non-methylated DNA. DNA from tumor tissue has a larger adsorption area on the gold surface than normal DNA (scale bar: 500 nm). (**C**) Atomic force microscope (AFM) images show methylated DNA has a higher adsorption cluster on the gold surface than non-methylated DNA. DNA from tumor tissue has a higher adsorption cluster on the gold surface than normal DNA (scale bar: 500 nm). (**D**) Left graph showed difference of binding groups between methylated DNA and non-methylated DNA adsorbed to gold surface by Fourier-transform infrared (FTIR). Right graph showed schematic of the potential mechanisms of differential adsorption capacity of methylated DNA and non-methylated DNA on the gold surface. * *p* < 0.05, *** *p* < 0.001.

**Figure 2 cancers-13-00664-f002:**
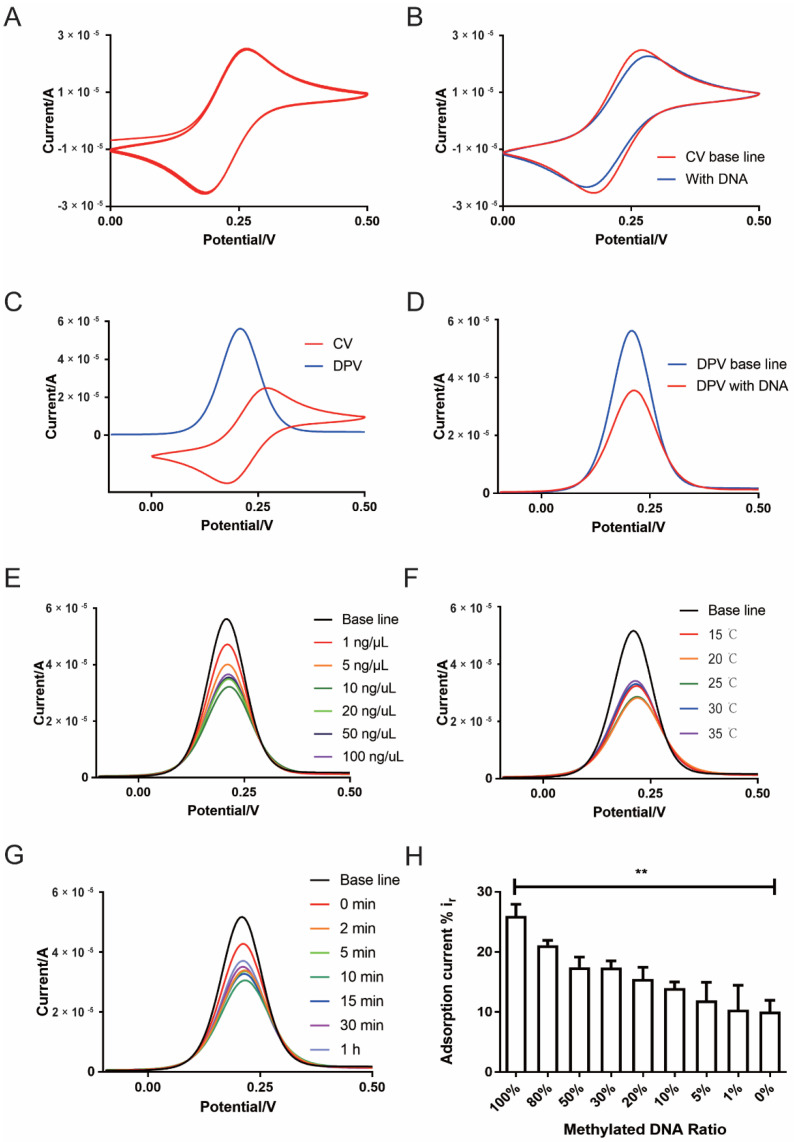
Establishment of an electrochemical detection method and exploration of the optimal detection condition. (**A**) The cyclic voltammetry (CV) curve showed that the system had better stability after the repeated testing of 100 segments. (**B**) The peak current of the CV curve decreased, and the potential difference increased after the adsorption of DNA. (**C**) The range of the differential pulse voltammetry (DPV) peak was determined by the oxidation peak of the CV curve in order to establish a more sensitive detection method. (**D**) The peak current of the DPV curve decreased significantly after DNA was adsorbed. (**E**–**G**) Optimization of the DNA concentration, the absorption temperature and the incubation time for the highest detection rate. (**H**) The current adsorption rate of methylated DNA in different proportions. ** *p* < 0.01.

**Figure 3 cancers-13-00664-f003:**
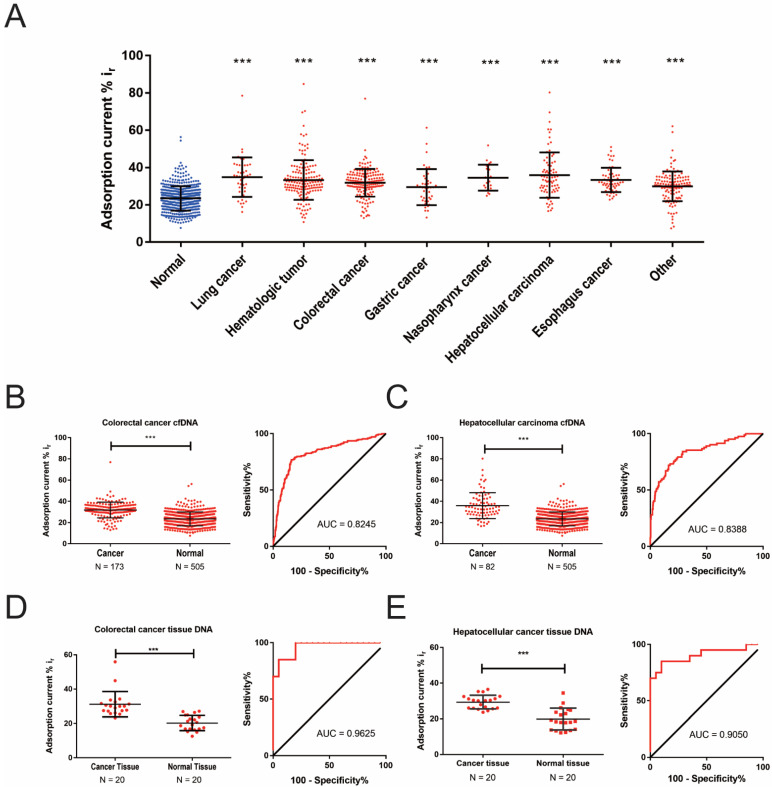
Application of electrochemical methylation detection in tissue and plasma circulating free DNA (cfDNA). (**A**) The scatter plot shows the electrochemical detection of the current adsorption rate of cfDNA from normal people, lung cancer, hematological cancer, colorectal cancer, gastric cancer, nasopharynx cancer, hepatocellular cancer, esophagus cancer and other cancers. (**B**) The scatter plot shows the current adsorption rate of cfDNA from colorectal cancer patients and normal people. Receiver operating characteristic (ROC) curve of the diagnostic efficacy of the cfDNA adsorption rate in colorectal cancer. (**C**) The scatter plot shows the current adsorption rate of cfDNA from hepatocellular cancer patients and normal people. ROC curve of the diagnostic efficacy of the cfDNA adsorption rate in hepatocellular cancer. (**D**) The scatter plot shows the current adsorption rate of tissue DNA from colorectal cancer and normal people. ROC curve of the diagnostic efficacy of the tissue DNA adsorption rate in colorectal cancer. (**E**) The scatter plot shows the current adsorption rate of tissue DNA from hepatocellular cancer and normal people. ROC curve of the diagnostic efficacy of the tissue DNA adsorption rate in hepatocellular cancer. *** *p* < 0.001.

**Figure 4 cancers-13-00664-f004:**
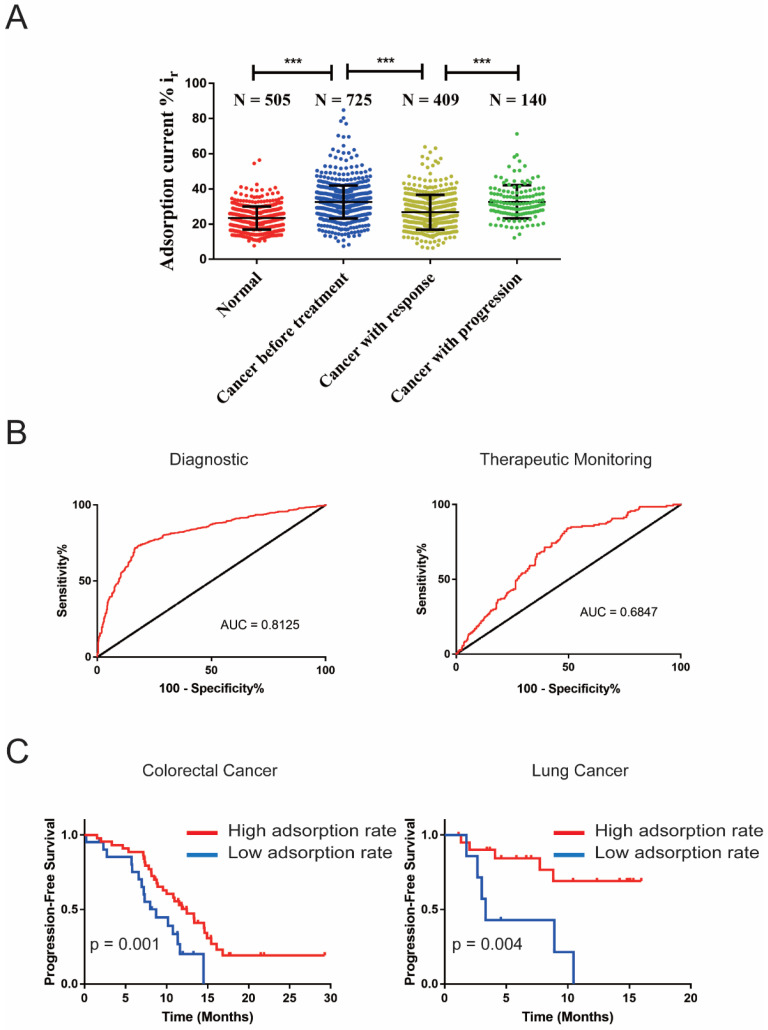
Application of electrochemical methylation detection in malignant tumor diagnosis and prognosis. (**A**) The scatter plot shows the electrochemical detection of the cfDNA adsorption rate from normal people, tumor patients before treatment, tumor patients with response after treatment and tumor patients with progression after treatment. (**B**) The ROC of the cfDNA adsorption rate in the diagnosis and recurrence detection of malignant tumor patients. (**C**) The survival curve shows the progression-free survival time of lung cancer and colorectal cancer patients with high adsorption rates and low adsorption rates. *** *p* < 0.001.

**Figure 5 cancers-13-00664-f005:**
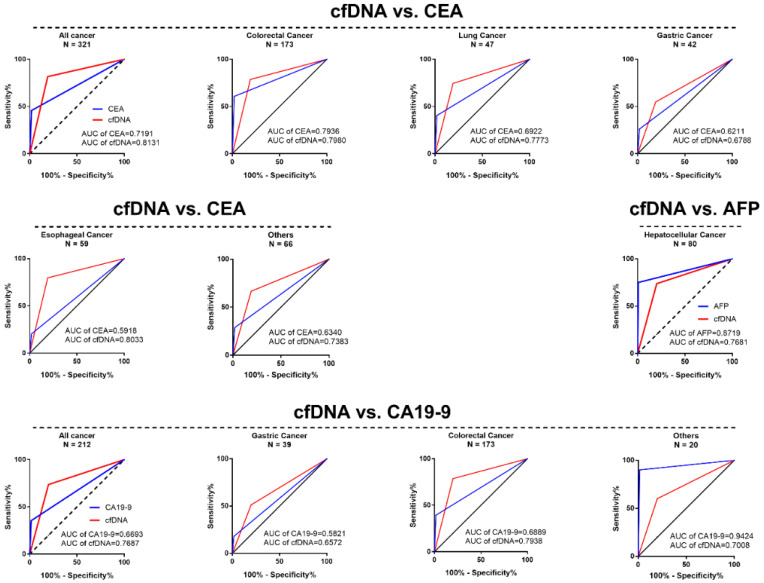
The ROC curve of the cfDNA adsorption rate compared with the tumor biomarkers carcinoembryonic antigen (CEA), CA19-9 and alpha fetoprotein (AFP) in the diagnosis of most malignant tumors.

**Table 1 cancers-13-00664-t001:** Clinical characteristics of plasma samples from normal people and patients with malignant tumors.

Variable	Normal	Cancer before Treatment	Cancer with Response	Cancer with Progression	*p*-Value
	(*n* = 505)	(*n* = 725)	(*n* = 409)	(*n* = 140)	
Age(year)	37.19 ± 12.06	52.64 ± 12.99	51.61 ± 13.48	53.16 ± 11.43	<0.001
Gender, *n* (%)					<0.001
Male	180(35.6)	457(63.0)	268(65.5)	100(71.4)	
Female	325(64.4)	268(37.0)	141(34.5)	40(28.6)	
tumor-node-metastasis (TNM) stage, *n* (%)					
I		107(14.8)			
II		142(19.6)			
III		187(25.8)			
IV		273(37.7)			
Unclearly		16(2.2)			
Differentiation status, *n* (%)					
Well-Moderate		243(36.1)			
Poorlyundifferentiated		210(31.2)			
Unclearly		272(32.7)			

## Data Availability

The data presented in this study are available in supplementary material.

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
