# Peer review of "Diagnostic and Prognostic Characteristics of Circulating Free DNA Methylation Detected by the Electrochemical Method in Malignant Tumors"

_cancers, 2021, doi:10.3390/cancers13040664_

Round 1
Reviewer 1 Report
Review Cancers -1084712
Title: Diagnostic and prognostic characteristics of circulating free DNA methylation detected by electrochemical method in malignant tumors
The manuscript entitled "Diagnostic and prognostic characteristics of circulating free DNA methylation detected by the electrochemical method in malignant tumors", by Li-Yue Sun et.al, established an electrochemical method based on the differential adsorption capacity of the gold surface with different degree of methylated DNA, and found that this method could be useful for cancer diagnosis by detecting methylation of circulating free DNA.
The manuscript appears to have improved; the authors found cfDNA methylation status from 505 normal individuals, 725 cancer patients before treatment and 549 patients after treatment, and found that the cfDNA adsorption rate could be used as an indicator for the diagnosis and prediction of the prognosis of pan-cancer.
By applying a highly sensitive and specific ctDNA sequencing assay on a cohort of 124 metastatic cancer patients and 47 controls without cancer, with matched white blood cell DNA, Razavi et al. found that 53.2% of mutations found in cancer patients had features consistent with clonal haematopoiesis. Razavi et al. highlights therefore the risk of false findings and the need to integrate white blood cell DNA as control when applying ultrasensitive ctDNA detection method. Overall, it appears necessary that laboratories should comment on these different limitations in their reports.
Razavi, P., Li, B. T., Brown, D. N., Jung, B., Hubbell, E., Shen, R. et al. High-intensity sequencing reveals the sources of plasma circulating cell-free DNA variants. Nat. Med. 25, 1928–1937 (2019).
On this way, I strongly suggest and invite to complete the manuscript with a predictive regression model to calculate the false positivity and negative rate of the method by comparing cfDNA methylation status with white blood cell DNAmethylation status in the same cancer patient. Even a group of few cases would be sufficient to suggest that an elegant methodology, such as the one presented by Li-Yue Sun, cannot be considered inappropriate because strongly compromised by the scarce specificity of the biomarker analysed.
Author Response
REVIEWER 1:
Comments and Suggestions for Authors
Review Cancers -1084712
Title: Diagnostic and prognostic characteristics of circulating free DNA methylation detected by electrochemical method in malignant tumors
The manuscript entitled "Diagnostic and prognostic characteristics of circulating free DNA methylation detected by the electrochemical method in malignant tumors", by Li-Yue Sun et.al, established an electrochemical method based on the differential adsorption capacity of the gold surface with different degree of methylated DNA, and found that this method could be useful for cancer diagnosis by detecting methylation of circulating free DNA.
The manuscript appears to have improved; the authors found cfDNA methylation status from 505 normal individuals, 725 cancer patients before treatment and 549 patients after treatment, and found that the cfDNA adsorption rate could be used as an indicator for the diagnosis and prediction of the prognosis of pan-cancer.
By applying a highly sensitive and specific ctDNA sequencing assay on a cohort of 124 metastatic cancer patients and 47 controls without cancer, with matched white blood cell DNA, Razavi et al. found that 53.2% of mutations found in cancer patients had features consistent with clonal haematopoiesis. Razavi et al. highlights therefore the risk of false findings and the need to integrate white blood cell DNA as control when applying ultrasensitive ctDNA detection method. Overall, it appears necessary that laboratories should comment on these different limitations in their reports.
Razavi, P., Li, B. T., Brown, D. N., Jung, B., Hubbell, E., Shen, R. et al. High-intensity sequencing reveals the sources of plasma circulating cell-free DNA variants. Nat. Med. 25, 1928–1937 (2019).
On this way, I strongly suggest and invite to complete the manuscript with a predictive regression model to calculate the false positivity and negative rate of the method by comparing cfDNA methylation status with white blood cell DNA methylation status in the same cancer patient. Even a group of few cases would be sufficient to suggest that an elegant methodology, such as the one presented by Li-Yue Sun, cannot be considered inappropriate because strongly compromised by the scarce specificity of the biomarker analysed.
Reply: Thank the reviewer for the comment. According to the reviewer's suggestion, we supplemented the research content (Supplementary Figure 4). We detected the adsorption rate of cfDNA and white blood cell DNA in 18 patients with malignant tumor. The results showed that the adsorption rate of cfDNA in patients was significantly higher than that of white blood cell DNA, which further proved that the method could be used to can detect the DNA methylation characteristics derived from tumors in cfDNA. page 7, line 171-174).
Reviewer 2 Report
The Authors have addressed all comments raised in the first tour of review. Therefore, I recommend the paper for publication.
Author Response
Thank the reviewer for the comment.
This manuscript is a resubmission of an earlier submission. The following is a list of the peer review reports and author responses from that submission.
Round 1
Reviewer 1 Report
Reviewer. Prof Natalia Malara
Review paper Cancers
Title: Diagnostic and prognostic characteristics of the electrochemical method detect circulating free DNA methylation in malignant tumours
The manuscript cancers-1004411 titled " Diagnostic and prognostic characteristics of the electrochemical method detect circulating free DNA methylation in malignant tumours
By Authors: Li-Yue Sun, Zi-Ming Du, Ting Wang, Yu-Ying Liu, Jian-Yong Shao" describes an electrochemical method based on the differential adsorption of gold surface and different methylated DNA.
The study is interesting but lacks some fundamental experimental steps to arrive at the conclusions, nevertheless, reported in the manuscript.
Here are some major issues:
- The authors correlate the differential adsorption of gold surface and different methylated DNA degree with clinical condition of absence or presence of tumour, without a preliminary study and assessment of a chemometric model built even only/or even on a simulated experimental scale.
- The statistical study showing the degree of absorption with the clinical phenotype is based on the use of chi square test while in these cases the PCA is more indicated to understand the discriminating entity present between the groups considered
- The clinical phenotype of reference is divided into healthy and cancerous. There is no clinical definition of healthy and the cancer patients are not considered in order of TNM. Although in the introduction the authors self-report this lack, the study cannot be considered complete without entering these data and processing them.
Minor issues
The language should be improved
In conclusion: In the current format, the article should not be published.
Reviewer 2 Report
Please see attached.
need how/why and nos pr %s everywhere

Reviewer 3 Report
The article is interesting and the data of potential clinical value.
Major points:
1) Did the authors used microdissection to isolate tumor tissue for paraffin block? If not, what was the percentage of normal tissue present in the specimens? Did a pathologist check for this? Please clarify this point indicating the criteria you used
2) The authors should better describe the potential clinical implications of their results